# Machine-learning-powered extraction of molecular diffusivity from single-molecule images for super-resolution mapping

Ha H. Park [1], Bowen Wang[1], Suhong Moon[2], Tyler Jepson [3] & Ke Xu [1,3]✉

While critical to biological processes, molecular diffusion is difficult to quantify, and spatial mapping of local diffusivity is even more challenging. Here we report a machine-learning-enabled approach, pixels-to-diffusivity (Pix2D), to directly extract the diffusion coefficient $D$ from single-molecule images, and consequently enable super-resolved $D$ spatial mapping. Working with single-molecule images recorded at a fixed framerate under typical single-molecule localization microscopy (SMLM) conditions, Pix2D exploits the often undesired yet evident motion blur, i.e., the convolution of single-molecule motion trajectory during the frame recording time with the diffraction-limited point spread function (PSF) of the microscope. Whereas the stochastic nature of diffusion imprints diverse diffusion trajectories to different molecules diffusing at the same given $D$, we construct a convolutional neural network (CNN) model that takes a stack of single-molecule images as the input and evaluates a $D$-value as the output. We thus validate robust $D$ evaluation and spatial mapping with simulated data, and with experimental data successfully characterize $D$ differences for supported lipid bilayers of different compositions and resolve gel and fluidic phases at the nanoscale.

[1] Department of Chemistry, University of California, Berkeley, CA 94720, USA. [2] Department of Electrical Engineering and Computer Sciences, University of California, Berkeley, CA 94720, USA. [3] QB3-Berkeley, University of California, Berkeley, CA 94720, USA. ✉email: xuk@berkeley.edu

Molecular diffusion underlies vital cellular processes[1–4]. The diffusion coefficient $D$, a metric of how fast molecules diffuse, is a function of both the molecular size and intracellular parameters such as viscosity and inter-molecular interactions. This correlation between diffusion and intracellular parameters has led to the use of $D$ as a reporter of intracellular (micro)environments to correlate dynamic properties with structures.

Many fluorescence microscopy techniques have been developed to probe diffusion at different spatiotemporal scales. Traditional approaches such as fluorescence recovery after photobleaching (FRAP)[4] and fluorescence correlation spectroscopy (FCS)[3,5–8] offer limited capabilities for spatial mapping, and often encounter background and calibration challenges when applied to biological samples. Single-particle tracking (SPT)[9–13] provides high spatial-resolution measurements for molecular diffusion, and with recent developments integrating photo-activation and related concepts from single-molecule localization microscopy (SMLM), has substantially increased the density of single molecules that can be tracked in the sample[9,11]. However, SPT aims at obtaining long single-particle tracks, from which the $D$-value of each particle is calculated. The need to track the same particle over many consecutive frames limits the application to photostable particles consistently bound to the target, e.g., lipid membranes, while also limiting the spatial mapping capabilities[9,11,14].

We recently developed single-molecule displacement/diffusivity mapping (SM$d$M), which focuses on detecting the transient motion/velocity of single molecules across tandem camera frames under frame-synchronized stroboscopic illumination[15]. By eliminating the need to track long trajectories, the approach works well for unbound fluorescent proteins in the cell[15] and dye molecules dynamically entering and leaving lipid membranes[16]. The accumulation of many transient displacements over different frames further enables local statistics[17] and $D$ mapping. The need for frame-synchronized stroboscopic illumination, however, limits the adaptation of this technique. Moreover, each molecule needs to be successfully captured in two consecutive frames and correctly paired to yield a useful measurement, limiting the possible throughput.

Here we report a strategy, pixels-to-diffusivity (Pix2D), to directly extract $D$-values from single-molecule images recorded under typical SMLM conditions. We reason that for diffusing molecules, images recorded at a fixed camera framerate are the convolution of their motion trajectories and the microscope point spread function (PSF), with the faster-moving molecules exhibiting stronger motion blurs. Whereas such information may not directly yield a meaningful $D$-value for each molecule due to the stochastic nature of diffusion, accumulating many molecules over different frames may enable spatial binning for local analysis[17] and thus $D$ mapping at the super-resolution level. Although previous work has extracted $D$ from single-molecule images[18–20], a global $D$-value is obtained for each sample without spatial mapping, and the model-based fitting approach is susceptible to experimental factors such as camera pixelation effects and backgrounds.

In this work, we seize the rising opportunities of modern machine-learning approaches to directly link single-molecule images to $D$-values without assuming any models, and we further achieve super-resolution mapping. Recent years have witnessed the fast growth of machine learning in single-molecule microscopy[21], with applications ranging from the enhancement of localization[22–33] and tracking[34–36] to the characterization of diffusive modes and properties[37–40]. In a previous study, we demonstrated the use of neural networks to connect single-molecule images to their color and depth information, thus enabling two-color three-dimensional SMLM[25]. Here we construct a convolutional neural network (CNN) model to connect stacks of single-molecule images to $D$-values. Spatially binning single-molecule images accumulated from many frames further enables local image stacks to be used as inputs for the model to generate $D$ maps at the super-resolution level.

## Result

### Construction of diffusivity-mapping CNNs

We constructed a CNN architecture that maps single-molecule images to $D$-values. As discussed, an individual single-molecule image does not provide a meaningful readout for $D$. Thus, we built a CNN model that took a stack of uncorrelated single-molecule images as the input and predicted a $D$-value as the output. Following the design principles of ResNet[41], a seminal work in deep neural network architectures, we designed the CNN model stacking multiple convolutional layers, a fully connected layer, and the final regression layer (Supplementary Fig. 1). Each convolutional building block entailed a convolutional layer with a kernel size of 3, a batch normalization layer[42], and Swish activation[43]. Whereas ResNet uses the ReLU[44] activation function, we employed the Swish activation function for our regression task. A relatively shallow CNN model was implemented to avoid overfitting.

Here we focus on diffusion in a two-dimensional system, so that single-molecule motion blurs are not further convoluted with off-focusing. For training, single-molecule images were generated by mapping simulated two-dimensional Brownian trajectories (Fig. 1a and Methods) to pixelized intensities using the PSF profile of our microscope setup (Fig. 1b). To match typical experimental settings, simulations were performed for a pixel size of 160 nm and an exposure time of 9 ms per frame. The input was a stack ($n_{ch}$ channels) of uncorrelated single-molecule images cropped at $7 \times 7$ pixels (Fig. 1c). Training (Fig. 1d) was performed on a dataset augmented with different molecule brightnesses and backgrounds (see below), with each diffusivity label containing hundreds of such stacks as the training inputs (Methods).

For spatial mapping of diffusivity, single-molecule images (Fig. 1e) were first localized in each frame via centroid fitting. A region of interest (ROI) of $7 \times 7$ pixels was sampled around the center of each localized molecule, and molecules with overlapping ROIs were discarded. The localized molecules, accumulated from many frames, were then spatially binned onto a fine grid (~100 nm) (Fig. 1f), so that each bin had a pool ($p^i$ for bin $i$) of single-molecule images. For every bin satisfying $p^i > 10$, $m = 100$ inputs, each with the same dimensionality as the training data—thus 100 different $7 \times 7 \times n_{ch}$ arrays—were generated by random sampling from the $p^i$ images for feeding into the model (Fig. 1g). The resultant 100 outputs from the model were averaged to yield the diffusivity of the bin. The results of different spatial bins were then color-coded to generate a spatial map (Fig. 1h).

### Performance of Pix2D

Figure 2a shows the model prediction results with $n_{ch} = 40$, using evaluation data not included in the training data. To avoid biases due to capping at the upper bound of the training range, here we trained the model with $D = 0–6\ \mu m^2/s$ data, and applied it to evaluate data generated with $D$ of $0–5\ \mu m^2/s$. Statistics of the Pix2D results on 400 evaluation inputs at each diffusivity yielded averages (Fig. 2a, black dots) well following the ground truth (Fig. 2a, red line), and the relative errors, $\%\mathrm{Error} = (E - T)/T\,S \times 100\%$, with $E$ and $T$ respectively being the Pix2D-evaluated and ground-truth $D$-values, showed typical standard deviations of ~12% (Fig. 2a shaded areas and Fig. 2b).

We next examined how the relative standard error depends on the data size. Comparing the model evaluation results with

different $n_{ch}$ and $p^i$ values in the range of 20-160 (Supplementary Fig. 2) indicated that the final precision depends only on the latter, namely, the count of starting single-molecule images. Thus, the choice of $n_{ch}$ was non-critical, and we found the $n_{ch} = 40$ and $m = 100$ combination we used practical. Plotting the relative standard errors as a function of $p^i$ showed a monotonic decrease for an increased single-molecule count (Fig. 2c). We have recently shown that with maximum likelihood estimation (MLE) based on the distribution of step distances, the relative standard error in $D$ equals to the inverse square root of the count of step distances[45]. For the same number of single-molecule images, the Pix2D standard errors were consistently ~2-fold lower (Fig. 2c). This behavior is reasonable, considering that in Pix2D, each single-molecule image directly encodes motion in two dimensions, whereas for approaches based on the analysis of step distances, each step needs two single-molecule images, and each connected single-step displacement is along one direction.

**Spatial diffusivity mapping of simulated data.** To examine the performances of Pix2D for diffusivity mapping, we simulated spatial patterns, e.g., square and circular shapes ~1 μm in size with contrasting regional $D$-values of 2 and 4 μm²/s (Fig. 2d, left). Single-molecule trajectories were simulated by randomly selecting the starting positions of the molecules and then updating $D$ at every micro-step based on their new positions. The simulated single molecules were localized and spatially binned into a 120 nm × 120 nm grid, so that the resultant counts of molecules were ~200 per bin, comparable to the typical experimental super-resolution imaging data. Single-molecule images in each bin were then processed and fed into the model described above for diffusivity mapping (Fig. 2d, right). We thus showed that our approach correctly mapped out spatial differences in diffusivity. Line profiles crossing the diffusion pattern boundaries showed sharp transitions in the mapped $D$-values (Fig. 2e), so that full transitions were accomplished within ~2 bins (~240 nm). Relative errors to the ground truths were <10% for most bins for the entire images (Fig. 2f).

**$D$ mapping of experimental data on supported lipid bilayers.** To apply the above model trained with simulated datasets to experimental single-molecule images, we first augmented the training data to deal with two potentially highly variable parameters, the brightness of molecules and background noise. We adapted domain randomization[46] to achieve performance invariant toward these two parameters. Multiple training datasets were generated from combinations of different photon counts and background noise levels when mapping the diffusion trajectories to images. The ranges of these parameters were set to mimic the typical distributions of single-molecule brightness and background of wide-field single-molecule images. The resultant model performed robustly for different combinations of parameters both covered (Supplementary Fig. 3) and not covered (Supplementary Fig. 4) in the augmentation.

Experimental single-molecule images were obtained with a typical SMLM setup for BDP-TMR-alkyne, which reversibly intercalated into supported lipid bilayers (SLBs) to report the lateral diffusivity of the membrane[16]. Single-molecule images (Fig. 3a) were collected with an exposure time of 9 ms per frame under typical SMLM conditions for SLBs prepared from pure 1,2-dioleoyl-*sn*-glycero-3-phosphocholine (DOPC) and mixtures of

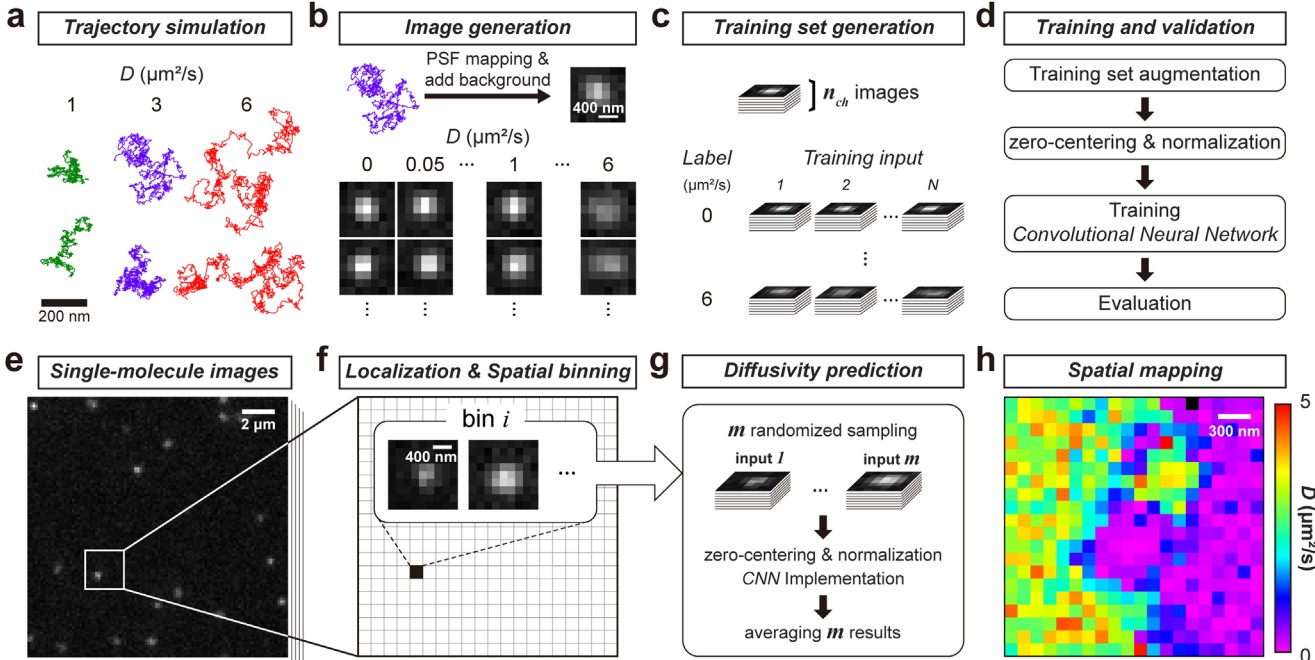

**Fig. 1 Pix2D CNN training and implementation. a** Examples of simulated Brownian trajectories for $D = 1$, 3, and 5 μm²/s for a camera exposure time of 9 ms. **b** Mapping of simulated trajectories to pixelized images using the microscope point spread function profile and a pixel size of 160 nm. 400 independent trajectories were initially generated for every $D$-value in the range of 0–6 μm²/s at a 0.05 μm²/s spacing. **c** As CNN training input, for each $D$ label, hundreds of stacks of $n_{ch}$ simulated images, each sampled as 7 × 7 pixels, were initially selected per diffusivity label. **d** The training data were then augmented *via* domain randomization, in which multiple image sets were created through identical processes as in **a–c**, but with different combinations of noise levels and photon counts. The CNN model was then trained with the augmented data. **e–h** Implementation of Pix2D diffusivity spatial mapping. Single-molecule signals from the raw image sequence were localized and spatially binned with a fixed grid size (**f** is a zoom-in of the boxed region in **e**). **g** For each bin, $m = 100$ permutations of $n_{ch}$ single-molecule images were each fed to the trained model. The resultant $m$ predicted $D$-values were averaged to give the diffusivity of the bin. **h** The results of different spatial bins were color-coded to generate a spatial map.

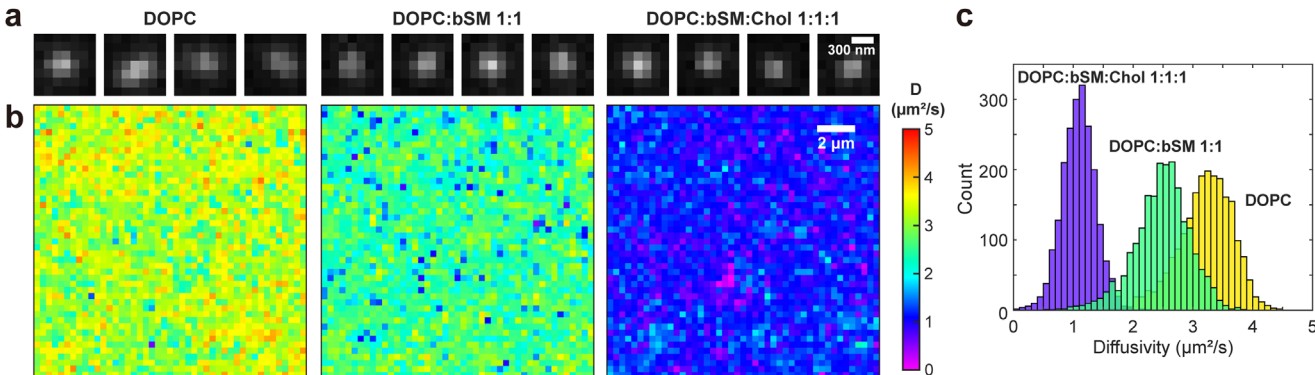

**Fig. 2 Performance assessment for Pix2D CNN. a** Validation of the trained CNN using simulated data not included in the training, for input channel size $n_{ch} = 40$. For each $D$-value, evaluations were performed for 400 sets of $n_{ch}$ images, and the average value and standard deviation were plotted as a black marker and the shaded area, respectively. Red line: reference (evaluation equaling the ground truth). **b** Distributions of relative errors (%Error) for training labels of $D = 2$ and $4\,\mu m^2/s$. **c** Standard deviations of %Error for $D = 1, 2, 3$, and $4\,\mu m^2/s$ for different counts of starting single-molecule images ($p^i$), but fixed $n_{ch} = 40$. Red dashed line: trend based on MLE analysis of the distribution of single-molecule step distances. **d** Spatial mapping of simulated data. (Left) Ground truths of simulated patterns with spatially varied $D$-values. (Right) Pix2D mapping results at a binning grid size of 120 nm, so that each bin counted ~200 simulated single-molecule images. **e** Line profiles in three rows of the Pix2D mapping results along the black boxes in **d**. Dashed line: reference line of ground truth diffusivity. **f** Distribution of the %Error of Pix2D results in the different spatial bins, for all bins in **d**.

**Fig. 3 Pix2D applied to experimental data on single molecules diffusing in supported lipid bilayers (SLBs). a** Example wide-field single-molecule images collected at an exposure time of 9 ms per frame, for BDP-TMR-alkyne diffusing in SLBs of different compositions: DOPC only, DOPC:bSM 1:1 (mol%), and DOPC:bSM:cholesterol 1:1:1 (mol%). **b** Color-coded $D$ maps of the three samples generated by spatially binning the single-molecule images accumulated over 30,000 frames onto a 320 nm × 320 nm grid, and then feeding the resultant ~200 single-molecule images in each bin into the Pix2D CNN for evaluation of local $D$. **c** Distributions of the evaluated $D$-values in each bin for the three samples, corresponding to 3.21 ± 0.45, 2.48 ± 0.46, and 1.09 ± 0.28 $\mu m^2/s$, respectively.

DOPC:bSM (brain sphingomyelin) 1:1 (mol%) and DOPC:bSM:cholesterol 1:1:1 (mol%). The localized single-molecule images were processed through Pix2D to generate diffusivity maps. Each spatially homogeneous, yet separately distinct diffusion coefficients were thus visualized for the three SLBs (Fig. 3b). Statistics of the evaluation results of different spatial bins yielded $D = 3.21 \pm 0.45$, $2.48 \pm 0.46$, and $1.09 \pm 0.28 \, \mu m^2/s$, respectively, for the three SLBs (Fig. 3c). These values are comparable to previously reported results, in which $D \sim 3\text{-}4$, $\sim 2\text{–}3$, and $\sim 1 \, \mu m^2/s$ are reported for similar DOPC, DOPC:bSM, and DOPC:bSM:cholesterol SLBs at room temperature[47,48]. We also performed single-particle tracking[12,13] on the data, but found most molecules only stayed in the SLB for a few frames (Supplementary Fig. 5), as is typical in PAINT-type SMLM[49]. Mean squared displacement (MSD) analysis on the occasionally observed long traces (Supplementary Fig. 6) yielded the same trends for the three SLBs, consistent with the notion that diffusion is slower in the more densely packed phases containing saturated lipids and cholesterol[47,48].

***D* mapping of microdomains in supported lipid bilayers**. To further assess mapping capabilities, we prepared SLBs with spatially separated domains. With a 60:40 mixture of DOPC and 1,2-dipalmitoyl-*sn*-glycero-3-phosphocholine (DPPC), we generated SLBs in which DOPC and DPPC respectively segregated into the fluid and gel phases at room temperature[50]. SMLM super-resolution images (Fig. 4a) of BDP-TMR-alkyne single molecules collected at an exposure time of 9 ms per frame showed contrasting localization densities for the DOPC and DPPC domains, as the higher packing order of aliphatic chains of the gel phase DPPC limited fluorophore access[51]. This difference in localization density disappeared upon melting of the DPPC domains at high temperatures (Supplementary Fig. 7). Binning the accumulated single-molecule images with a 120 nm × 120 nm grid allowed mapping of local $D$ with Pix2D (Fig. 4b). Contrasting $D$ was thus unveiled for the two lipid phases. Temporally dividing the collected single-molecule data into two periods of 3.4-min durations for their separate Pix2D evaluations further yielded $D$ maps comparable to the entire dataset (Supplementary Fig. 8), thus demonstrating that the diffusivity spatial patterns remained unchanged over the recording time.

Comparison of the Pix2D $D$ map (Fig. 4b) with the SMLM image (Fig. 4a) showed good correlations: The high-count regions (DOPC phase) consistently exhibited high $D$ of $\sim 3.2 \, \mu m^2/s$, close to our above results on the single-component DOPC SLB (Fig. 3). Meanwhile, the low-count regions (DPPC phase) exhibited $D < \sim 0.5 \, \mu m^2/s$, consistent with the notion that this gel phase is

nonfluidic at room temperature. Notably, features down to $\sim 300$ nm were well-resolved in the Pix2D $D$ map (Fig. 4c), in agreement with our simulation results (Fig. 2d, e). Distribution of the evaluated local $D$-values in each spatial bin versus the count of molecules in the bin further showed a good correlation that fast diffusion was exclusively observed for the high-count DOPC phase (Fig. 4d).

## Discussion

While critical to biological processes, molecular diffusion has been difficult to quantify, and spatial mapping of local diffusivity is even more challenging. In this work, we exploited the often undesired yet evident motion blur of single-molecule images, recorded under typical SMLM conditions, to extract diffusion coefficient $D$ and further enable spatial mapping. Many recent efforts have successfully applied machine learning to single-molecule images in SMLM contexts to assist three-dimensional localization and color separation, as well as to extract optical parameters such as the Zernike coefficients and fluorescent backgrounds[21–33] In this study, we instead focused on extracting higher-dimensional, functional information[52] on molecular diffusion from the single-molecule images.

Different from previous machine-learning studies in which the parameters in question are directly projected to single-molecule images with a fixed pattern, the stochastic nature of diffusion mandates that even for a fixed $D$-value, the diffusion trajectory of a molecule in a fixed time window (and thus motion blur) takes diverse forms not unique to the given $D$. Thus, rather than attempting to assign a $D$-value to each single-molecule image, we constructed a model that took a stack ($n_{ch} \sim 40$ channels) of uncorrelated single-molecule images as the input and evaluated a $D$-value as the output. For input data of different numbers of single-molecule images, $m \sim 100$ sets of $n_{ch}$ samplings were separately fed into the model, and the averaged output of the $m$ datasets was taken as the final $D$-value. This model architecture of fixed channel numbers simplified implementation, while the $m$-time sampling provided flexibility for the number of source single-molecule images. These advantages proved instrumental to spatial mapping: we thus were able to spatially grid all the collected single-molecule images and separately fed the images in each spatial bin to the CNN model to evaluate local $D$, without having to worry about the different counts of molecules in each bin.

With simulated data of known ground truths, we thus showed that our above Pix2D approach correctly extracted $D$ from single-molecule image stacks. With 40 source single-molecule images, the typical relative standard error $\sigma_{error}$ was $\sim 12\%$ over a wide $D$

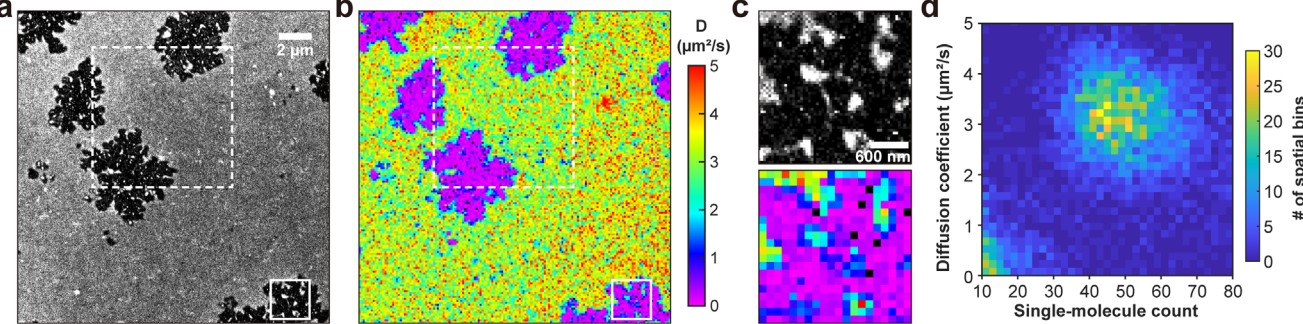

**Fig. 4 Pix2D *D* mapping of SLB microdomains. a** SMLM super-resolution image of an SLB of 60:40 DOPC:DPPC, presented as the local counts of single BDP-TMR-alkyne molecules recorded over 45,000 frames at 110 frames per second with an exposure time of 9 ms per frame. **b** Color-coded $D$ map generated by spatially binning the single-molecule images onto a 120 nm × 120 nm grid for the Pix2D evaluation of local $D$. **c** Zoom-in of the solid box-marked region in **a**, **b**. **d** Two-dimensional distribution of the Pix2D-evaluated $D$ versus single-molecule count for the different spatial bins in the dashed box-marked region in **a**, **b**.

range, and the evaluation results were robust towards different brightness and background noise levels. This evaluation error was ~2-fold better than that based on the MLE analysis of single-molecule step distances, and decreased monotonically for increased numbers of single-molecule images with little dependence on $n_{ch}$.

With experimental data on BDP-TMR-alkyne single molecules that dynamically entered SLBs and stayed for short durations, we next showed that Pix2D successfully resolved the different diffusivities for bilayers of different lipid compositions, with the resulting $D$-value trends matching that expected based on previous bulk measurements. For the phase-separated DOPC-DPPC mixture system, we further demonstrated that Pix2D resolved the different $D$-values in the two phases at the nanoscale.

Together, by directly linking stacks of uncorrelated single-molecule images to diffusivity, we have successfully extracted $D$-values from both simulated and experimental data, and further achieved spatial mapping under SMLM settings. Whereas with phase-separated SLBs we have resolved spatial patterns in the environment, the capability of detecting local diffusivities may also be harnessed to report on changes in the states of the diffusers themselves, e.g., oligomerizations and conformational changes. While in this work we have focused on the direct extraction of diffusivity from single-molecule images in individual frames, future efforts may consider connecting molecules spanning consecutive frames to further enhance prediction. The possible extension of the approaches developed in this work to diffusion in three dimensions, as well as to other high-dimensional single-molecule signal dimensions[17], represent additional exciting challenges.

## Methods

**Data simulation.** For training and test data, single-molecule images were generated from Monte Carlo two-dimensional Brownian diffusion simulation. For each molecule, a Brownian trajectory of 9 ms was simulated through 1000 micro-steps of random two-dimensional motion of distance $\sqrt{4Dt}$ using the local diffusion coefficient $D$ and the simulation interval $t$. The simulated trajectory was mapped onto the pixelized image grid using typical photon counts of 60-120 photons per ms and the microscope PSF profile experimentally determined using 100-nm fluorescence beads. Background noise was further added to each pixel based on the typical values observed with the recording camera.

**Model architecture, training, and application.** The Pix2D CNN model was designed to take a stack ($n_{ch}$ channels) of single-molecule images, each cropped at $7 \times 7$ pixels, as the input, and predict a $D$-value as the output. The architecture stacked multiple convolutional layers, a fully connected layer, and the final regression layer (Supplementary Fig. 1). The final prediction was computed by a fully connected layer, followed by a regression layer that calculated the mean square error (MSE) losses between the predicted values and the ground truth labels. Design methodologies of ResNet[41] were partially followed, so that the building blocks entailed a convolutional layer with a kernel size of 3, Swish activations[43], and batch normalization layers[42]. As we performed regression for predicting continuous $D$-values, we used Swish activation instead of the ReLU activation used in the original ResNet model. The training dataset was constructed for 121 labels of diffusion coefficients, from 0 to 6 $\mu m^2$/s in 0.05 $\mu m^2$/s intervals. For each diffusion coefficient label, 100 training inputs of $n_{ch} = 40$ simulated $7 \times 7$ single-molecule images were used. Therefore, the training dataset had 12,100 inputs of the dimension of $7 \times 7 \times 40$, for which we further augmented 20 combinations of single-molecule brightnesses and image backgrounds. Further increasing the training dataset size did not further improve the prediction accuracy. The training objective was the MSE loss between the predicted diffusivity and the target diffusivity using stochastic gradient descent (SGD) with momentum as the optimizer. Training was performed over 60 epochs with a batch size of 128, and an initial learning rate of 0.003. The learning rate was decreased by 10x after every 25 epochs. To apply the trained CNN model, $m = 100$ datasets, each with the same dimensionality as the training data—thus $m$ different $7 \times 7 \times n_{ch}$ arrays—were generated by random sampling from the $p^i$ input images for feeding into the model. For bins having $p^i < n_{ch}$, each input is generated by allowing repeated sampling of each image, yet limiting the repetition to the minimum integer $j$ satisfying the following condition: $j \times p^i > n_{ch}$. The resultant $m$ outputs from the model were averaged to yield the final diffusivity. Negative final values were treated as zero for physical relevance.

**SLB preparation.** Lipids (Avanti Polar Lipids 850375, 850355, and 860062) and cholesterol (Sigma, C8667) were dissolved in chloroform as 5 mg/mL stock solutions. The lipid mixture was combined in a 25-mL round bottom flask with the desired ratio from the stock solution. The solvent was removed under a stream of nitrogen gas. The resulting lipid film was rehydrated in 60 °C Milli-Q water and vortexed to a final concentration of 1 mg/mL multi-lamellar vesicle (MLV) solution. The MLV solution was then extruded at 60 °C over 11 times through a 100 nm polycarbonate membrane filter (Avanti Mini Extruder) to form small unilamellar vesicles (SUVs). The SUV solution was diluted in a 2:1:1 mixture of PBS (phosphate-buffered saline):H$_2$O:SUV and sonicated until deposited on a piranha-etched coverslip. Excess, unruptured vesicles were removed after 20 min by washing with PBS. 1 nM of BDP-TMR-alkyne (A24B0, Lumiprobe) in PBS was used for the single-molecule imaging of SLBs.

**Single-molecule imaging.** Fluorescence imaging was performed on a Nikon Ti-E inverted fluorescence microscope. A 561 nm laser (OBIS 561 LS, Coherent, 165 mW) was focused at the back focal plane of an oil-immersion objective lens (Nikon CFI Plan Apochromat λ 100×, numerical aperture 1.45) to continuously illuminate the sample at ~1 kW/cm$^2$. A translation stage shifted the laser beams toward the edge of the objective lens, so the light reached the sample at an incidence angle close to the critical angle of the glass-water interface to achieve a near-total internal reflection condition. Wide-field single-molecule images were filtered by a long-pass filter (ET575lp, Chroma) and a band-pass filter (ET605/70 m, Chroma), and recorded continuously using an EM-CCD camera (iXon Ultra 897, Andor) in the frame-transfer mode at 110 frames per second (fps) with a frame integration time of 9 ms. The typically recorded photon counts were 500–1200 for each BDP-TMR-alkyne molecule. 30,000-50,000 frames were typically recorded for each sample.

**Processing of single-molecule images.** Single molecules were first identified and localized with established methods using Insight3 (Dr. Bo Huang at University of California, San Francisco, and Dr. Xiaowei Zhuang at Harvard University), yielding typical densities of ~0.1 molecules/μm$^2$/frame. An ROI of $7 \times 7$ pixels was sampled around the center of each localized molecule, and molecules with overlapping ROIs were discarded. The single-molecule images were then spatially binned with a fixed grid size (e.g., 120 nm × 120 nm). For bins having single-molecule count $p^i > 10$, the accumulated single-molecule images were fed into the trained Pix2D model as described above for the evaluation of local $D$.

**Statistics and reproducibility.** SLBs of all compositions were replicated multiple times, and each sample was imaged for at least three different field-of-views to confirm global consistency of each condition. Data shown in the current study were arbitrarily selected among replicates.

**Reporting summary.** Further information on research design is available in the Nature Portfolio Reporting Summary linked to this article.

## Data availability

The datasets generated and analyzed during the current study are available from the corresponding author upon reasonable request. The source data behind the graphs in the paper can be found in Supplementary Data 1.

## Code availability

The codes for Pix2D diffusivity spatial mapping are available online: https://github.com/ha-park/Pix2D-NN-diffusivity-mapping.

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

## Acknowledgements

We thank Aaron Ghrist for initial efforts. This work was supported by the National Science Foundation (CHE-1554717 and CHE-2203518). H.H.P. and S.M. acknowledge support from Korea Foundation for Advanced Studies.

## Author contributions

H.H.P., B.W., and S.M. developed the Pix2D and carried out simulations. H.H.P. and T.J. prepared lipid membrane samples and performed imaging. H.H.P. analyzed the data. K.X. supervised research.

## Competing interests

The authors declare no competing interests.
