## [Peer Review File · Communications Biology]

Reviewers' comments:

Reviewer #1 (Remarks to the Author):

This work presents a machine learning (ML) method able to compose an spatial mapping of the diffusion coefficient of particles tracked by means of single-molecule localization microscopy. It presents both benchmarking results on simulated data as well as the application of the method to various experimental datasets. In general, I find the paper well written and convincing in its arguments. The method presented is very well presented and could be easily reproduced. Moreover, the authors provide Matlab codes for such. To note, the work builds strongly upon a method previously proposed by some of the authors (Ref 14), but extends it by means of ML techniques.

Here are some comments about the text. Of most importance are those related to the technical presentation of the ML method. I think that, after considering these, the work is suitable for publication.

1. The introduction covers very well the recent advances in the field, both in terms of the experimental approaches to study diffusion in biological environments, as well as numerical methods to assess their diffusive properties. The authors reference many works using ML in single molecule microscopy. However, I find that many of these references are mixed even if they treat different problems (e.g. localization, tracking, characterization of diffusive properties,...). It may be helpful for the reader to extend that sentence and separate these different applications. In particular, for the latter (characterizing anomalous diffusion with ML), I would suggest more recent references, e.g. [A] collects and benchmarks multiple methods as the ones referenced in Refs. 35 and 36. Moreover, [B] and [C] perform similar spatial maps and may be worth highlighting them.

2. Both in the abstract and the conclusion it is stated that "...Pix2D exploits the often undesired yet evident motion blur...". However, I can't find any proof backing up this argument in the results. I do not say that the CNN is learning to get information from this, but how can the authors be sure?

3. I also have concerns with the following, present too in the abstract and conclusion: "... the stochastic nature of diffusion mandates that even for a fixed D value, the motion pattern of a molecule in a fixed time window takes diverse forms not unique to the given D ". While I more or less understand the message the authors want to give, I think it is theoretically wrong. Given a D , the motion of a particle is fully characterized by such value. Having a short sample of the trajectory causes having poor statistics and maybe incorrectly predicting D . But these are not 'diverse motion patterns'. I would suggest the authors reviewing such sentences both in the abstract and in the conclusion.

4. The authors claim that ReLU activation is unsuitable for regression tasks. That is too strong of a claim, which should either be backed up by a proper reference or softened. In general, we can agree that other activations will work better, but that does not make ReLU unsuitable.

5. The authors express that "Training (Fig. 1d) was performed for each diffusivity label...". This seems to imply that different models were trained for different labels, which I guess is not the case? Also, it is not clear in the text that the diffusivity label equals the diffusion coefficient.

6. My major concern is the presentation of the training set and the definitions of p_i , m . I would suggest the authors to improve their description of these parameters.

7. Partially due to the latter, it is not clear to me what the size of the training dataset is. The authors state that the datasets are smaller than CIFAR and ImageNet, but having a number would be more useful to reproduce the work. Also, does the accuracy of the method improve by growing the training dataset?.

8. Could the authors extend on why they do evaluation only with $D_{\max} = 5$ but for instance do consider always the same D_{\min} ?

9. In Fig. 2b: i) how is the % Error calculated? Does a positive value mean overestimation or the opposite? In any case, the method does seem to over/sub estimate the error in some cases (e.g. $D = 4$). Can the authors comment on this?

10. It is not clear what happens when two particles are found in the same crop. It is said that for the benchmark "...molecules with their nearest neighbor closer than the signal-sampling size were discarded...". First, how is signal-sampling defined? Does this mean that only one particle will be found in each cropped image? This does not seem completely realistic for some of the experimental datasets considered. How can you deal with this?

11. The work considers that the spatial maps are static, but in such environments we can expect them to change. Have the authors considered such phenomena? Moreover, the authors consider that the diffusive properties emerge from the environment. However, there are many examples in which these changes occur because of the internal state of the particles. Is there any way of studying this effect with the presented method?

[A] "Objective comparison of methods to decode anomalous diffusion." Nature communications 12.1 (2021): 1-16.

[B] "InferenceMAP: mapping of single-molecule dynamics with Bayesian inference." Nature methods 12.7 (2015): 594-595.

[C] "Geometric deep learning reveals the spatiotemporal fingerprint of microscopic motion." arXiv preprint arXiv:2202.06355 (2022).

Reviewer #2 (Remarks to the Author):

In this manuscript by Park et al., the authors train a neural net that reports on diffusion coefficients, throughout the field-of-view, from single molecule diffusion movies. The main novelty seems to be the use of a neural net on single-image motion blur to extract D per x - y bin. Overall the method seems to produce reasonable results. I suggest to address the following points to improve the manuscript. The main points relate to clarity of representation, comparison with SOTA, and incorporating temporal information.

1. There is one basic question that comes up while reading the paper – even if there is information in a single molecule frame on D (and thanks to motion blur, this is a very reasonable assumption), wouldn't it make sense to also use the trajectory of the molecule? E.g. the distribution of spatial steps? (this point would come up in the next questions as well).

2. Unfortunately, there is no comparison with existing methods. For example – what does MSD or similar analysis on the experimental data look like?

3. To clarify – does each stack corresponds to a trajectory of a single molecule?

4. For each stack – the assumption is that there is no photobleaching of the molecule?

5. "molecules with their nearest neighbor closer than the signal sampling size were discarded". What is the definition of "sampling size"? does that mean 2 molecules in the same 7×7 bin?

6. What does "randomized sampling" mean (Fig 1g)? are the different frames within a single trajectory being "mixed"? is this what "permutations" (Fig 1 caption) refers to? Again, this is confusing because obviously there is information in the temporal order of the frames.

7. If I understand correctly, n_{ch} is the number of frames per trajectory that is being used. If so, I suggest to replace the word "channels" with "frames", which to me seems clearer.

8. "For every bin satisfying $p_i > 10$, $m = 100$ datasets, each with the same dimensionality as the training data – thus m different $7 \times 7 \times n_{ch}$ arrays – were generated by random sampling from the p_i

images for feeding into the model (Fig. 1g)."

Just to be clear - Does this mean that if a bin has, for example, only 11 images of a single molecule, these 11 images are replicated, in random permutations, to generate 4000 ($m=100$ times $n_{ch}=40$) images, which are fed to the net m times in batches of size n_{ch} ?

9. Please add more detail on the final "regression layer" in the network.

10. It seems (Supp figure 1) that there are no skip connections, and the network is not very deep, therefore citing "Resnet" might not be the best fit.

11. Fig 2 c - the red line is not dashed, unlike what the caption says.

12. The SLB experimental results are said to be "consistent with the notion that diffusion is slower in more densely packed phases", but there is no quantitative comparison to past results. Can this be obtained?

13. In the SLB experiment - How long does a typical single molecule trajectory last before bleaching?

14. Figure 4d is not clear - what does "single-molecule counts" (x-axis) mean, vs. "Counts" in the color-bar?

15. Regarding future work - It seems to me that the most immediate extension of this work would be to combine the motion blur information with step-distance information, could the authors elaborate on what that would require?

Reviewer #1 (Remarks to the Author):

This work presents a machine learning (ML) method able to compose an spatial mapping of the diffusion coefficient of particles tracked by means of single-molecule localization microscopy. It presents both benchmarking results on simulated data as well as the application of the method to various experimental datasets. In general, I find the paper well written and convincing in its arguments. The method presented is very well presented and could be easily reproduced. Moreover, the authors provide Matlab codes for such. To note, the work builds strongly upon a method previously proposed by some of the authors (Ref 14), but extends it by means of ML techniques.

Here are some comments about the text. Of most importance are those related to the technical presentation of the ML method. I think that, after considering these, the work is suitable for publication.

Response: We thank the reviewer for his/her concise summary of our work and kind comments. Please see our point-by-point response below.

1. The introduction covers very well the recent advances in the field, both in terms of the experimental approaches to study diffusion in biological environments, as well as numerical methods to assess their diffusive properties. The authors reference many works using ML in single molecule microscopy. However, I find that many of these references are mixed even if they treat different problems (e.g. localization, tracking, characterization of diffusive properties,...). It may be helpful for the reader to extend that sentence and separate these different applications. In particular, for the latter (characterizing anomalous diffusion with ML), I would suggest more recent references, e.g. [A] collects and benchmarks multiple methods as the ones referenced in Refs. 35 and 36. Moreover, [B] and [C] perform similar spatial maps and may be worth highlighting them.

Response: Thank you for the discussion. We have revised our discussion to separate the different ML applications (Page 3, Paragraph 2): “with applications ranging from the enhancement of localization²²⁻³³ and tracking³⁴⁻³⁶ to the characterization of diffusive modes and properties³⁷⁻⁴⁰”. We have also added the references suggested by the reviewer, including the recently published version of [C].

2. Both in the abstract and the conclusion it is stated that “...Pix2D exploits the often undesired yet evident motion blur...”. However, I can’t find any proof backing up this argument in the results. I do not say that the CNN is learning to get information from this, but how can the authors be sure?

Response: We clarify that as we work with samples confined in two dimensions at the focal plane, the recorded single-molecule images correspond to the microscope point spread function (PSF) for stationary molecules. For moving molecules, the motion trajectory will be convoluted with the PSF to generate the final images recorded by the camera, which we refer to as “motion blur”. Our training data were exactly generated this way (Figure 1ab), and we have shown it worked well with experimental data, e.g., resolving the different D values in SLBs of different compositions and phases in Figures 3 and 4. We have improved our explanations. Abstract: “Pix2D exploits the often undesired yet evident motion blur, *i.e.*, the convolution of single-molecule motion trajectory during the frame recording time with the diffraction-limited point spread function (PSF) of the microscope.” Page 2, Paragraph 4: “We reason that for diffusing molecules, images recorded at a fixed camera framerate are the convolution of their motion trajectories and the microscope point spread function (PSF)”.

3. I also have concerns with the following, present too in the abstract and conclusion: “... the stochastic nature of diffusion mandates that even for a fixed D value, the motion pattern of a molecule in a fixed time window takes diverse forms not unique to the given D”. While I more or less understand the message the authors want to give, I think it is theoretically wrong. Given a D, the motion of a particle is fully characterized by such value. Having a short sample of the trajectory causes having poor statistics and maybe incorrectly predicting D. But these are not ‘diverse motion patterns’. I would suggest the authors reviewing such sentences both in the abstract and in the conclusion.

Response: As noted above, the recorded image of each molecule is a convolution of its trajectory and the microscope PSF. As the motion of each molecule is random, no two trajectories will be identical, even for molecules with the same D value. We realize that the wording ‘motion pattern’ may be unclear, and have changed it to “diffusion trajectory”. The two related sentences now read: “the stochastic nature of diffusion imprints diverse diffusion trajectories to different molecules diffusing at the same given D ” and “the diffusion trajectory of a molecule in a fixed time window takes diverse forms”.

4. The authors claim that ReLU activation is unsuitable for regression tasks. That is too strong of a claim, which should either be backed up by a proper reference or softened. In general, we can agree that other activations will work better, but that does not make ReLU unsuitable.

Response: Thank you for this discussion. We agree that although Swish activation is more commonly used for regression tasks, ReLU is still useable for related purposes. We have revised this discussion to remove the comment that ReLU is unsuitable (Page 3, Paragraph 3): “Whereas ResNet uses the ReLU⁴² activation function, we employed the Swish activation function for our regression task.”

5. The authors express that “Training (Fig. 1d) was performed for each diffusivity label...”. This seems to imply that different models were trained for different labels, which I guess is not the case? Also, it is not clear in the text that the diffusivity label equals the diffusion coefficient.

Response: Thank you for this question. Indeed, for the different diffusivity labels, we meant different diffusion coefficients, and they were all trained under the same model. We have clarified our description (Page 4, Paragraph 1): “Training (Fig. 1d) was performed on a dataset augmented with different molecule brightnesses and backgrounds (see below), with each diffusivity label containing hundreds of such stacks as the training inputs (Methods).”

6. My major concern is the presentation of the training set and the definitions of p_i , m . I would suggest the authors to improve their description of these parameters.

Response: We have improved our description (Page 4, Paragraph 2): “ $m = 100$ inputs, each with the same dimensionality as the training data – thus 100 different $7 \times 7 \times n_{ch}$ arrays – were generated by random sampling from the p^i images for feeding into the model (Fig. 1g). The resultant 100 outputs...”

7. Partially due to the latter, it is not clear to me what the size of the training dataset is. The authors state that the datasets are smaller than CIFAR and ImageNet, but having a number would be more useful to reproduce the work. Also, does the accuracy of the method improve by growing the training dataset?.

Response: Thank you for this comment. We have addressed these questions in our revision (Page 9, Paragraph 3): “The training dataset was constructed for 121 labels of diffusion coefficients, from 0 to 6 $\mu\text{m}^2/\text{s}$ in 0.05 $\mu\text{m}^2/\text{s}$ intervals. For each diffusion coefficient label, 100 training inputs of $n_{ch} = 40$ simulated 7×7 pixels single-molecule images were used. Therefore, the training dataset had 12,100 inputs of the dimension of $7 \times 7 \times 40$, for which we further augmented 20 combinations of single-molecule brightnesses and image backgrounds. Further increasing the training dataset size did not further improve the prediction accuracy.”

8. Could the authors extend on why they do evaluation only with $D_{\text{max}} = 5$ but for instance do consider always the same D_{min} ?

Response: Thank you for this discussion. Physically, D_{min} can only be as low as zero (no motion). Thus in our training, labels of diffusion coefficients have 0 as the minimum. The output values, on the other hand, contained negative values as the model performed a regression task. Negative final diffusivity values were treated as zero for physical relevance. We have added this information in the Methods section (Page 10, Paragraph 1).

9. In Fig. 2b: i) how is the % Error calculated? Does a positive value mean overestimation or the opposite? In any case, the method does seem to over/sub estimate the error in some cases (e.g. $D = 4$). Can the authors comment on this?

Response: The % Error is $(E - T)/T \times 100\%$, where E is the evaluation diffusion coefficient and T is the true diffusion coefficient. Thus, a positive value means the result is overestimated and *vice versa*. The distributions of errors were generally central toward 0. The average values of the evaluated D are plotted as black dots in Figure 2a: Scattering from the true values (red line in Figure 2a) was due to the stochastic nature of sampling. We have improved our discussion (Page 4, Paragraph 3): “Statistics of the Pix2D results on 400 evaluation inputs at each diffusivity yielded averages (Fig. 2a, black dots) well following the ground truth (Fig. 2a, red line), and the relative errors, %Error = $(E - T)/T \times 100\%$, with E and T respectively being the Pix2D-evaluated and ground-truth D values, showed typical standard deviations of $\sim 12\%$ (Fig. 2a shaded areas and Fig. 2b).”

10. It is not clear what happens when two particles are found in the same crop. It is said that for the benchmark “...molecules with their nearest neighbor closer than the signal-sampling size were discarded...”. First, how is signal-sampling defined? Does this mean that only one particle will be found in each cropped image? This does not seem completely realistic for some of the experimental datasets considered. How can you deal with this?

Response: Yes, in our analysis we ensure that only one particle is found in each cropped image. Single-molecule localization microscopy (SMLM) works under the premise that in each frame, the images of different single molecules are well separated to enable their independent localization. In our experiments, we work with typical single-molecule densities of ~ 0.1 molecules/ μm^2 /frame. Thus, with the typical single-molecule image size being $\sim 0.1 \mu\text{m}^2$, the chance for two single-molecule images to overlap is quite low. In our data processing, we use typical SMLM software to first identify and localize all single molecules. 7×7 pixels are sampled around the center of each localized molecule, and any overlapped sampling was rejected. As we recorded at 110 frames per second as in typical SMLM, sufficient counts of single-molecule images were accumulated in minutes; see also our answer below re #11. We have improved related discussion in the text. (Page 4, Paragraph 2): “A region of interest (ROI) of 7×7 pixels was sampled around the center of each localized molecule, and molecules with overlapping ROIs were discarded.” (Page 10, Paragraph 4): “... yielding typical densities of ~ 0.1 molecules/ μm^2 /frame.”

11. The work considers that the spatial maps are static, but in such environments we can expect them to change. Have the authors considered such phenomena? Moreover, the authors consider that the diffusive properties emerge from the environment. However, there are many examples in which these changes occur because of the internal state of the particles. Is there any way of studying this effect with the presented method?

Response: Thank you for this insightful question. Indeed, to map diffusivity, we assume local D values invariant over the measurement time. The same is generally true for SMLM, as super-resolution images emerge from the accumulation of single-molecule localizations over time. The experimental systems we examined in this work did not exhibit spatial changes over our measurement times. To show this, we perform a new analysis in our new Supplementary fig. 8, in which we divide the original single-molecule data of Figure 4b collected over 6.8 min into two periods of 3.4-min duration each. A larger spatial bin of $160 \text{ nm} \times 160 \text{ nm}$ grid was used for the Pix2D evaluation of local D , to ensure sufficient molecule counts in each spatial bin, and we obtained D maps comparable to Figure 4b for both periods. This analysis demonstrates that the diffusion spatial patterns indeed remained unchanged over our recording time, while also showing that viable D maps can be obtained at ~ 3 min time resolution. We have added the above discussion to the text (Page 7, Paragraph 1) and Supplementary fig. 8.

We agree that as our measurement focuses on detecting single-molecule motions and mapping diffusivity, it could report on changes in both the environment and the particles themselves. We have added this discussion to the text (Page 8, Paragraph 4): “Whereas with phase-separated SLBs we have resolved spatial

patterns in the environments, the capability of detecting local diffusivities may also be harnessed to report on changes in the states of the diffusers themselves, e.g., oligomerizations and conformational changes.”

[A] "Objective comparison of methods to decode anomalous diffusion." Nature communications 12.1 (2021): 1-16.

[B] "InferenceMAP: mapping of single-molecule dynamics with Bayesian inference." Nature methods 12.7 (2015): 594-595.

[C] "Geometric deep learning reveals the spatiotemporal fingerprint of microscopic motion." arXiv preprint arXiv:2202.06355 (2022).

Response: These references were for Question #1. Please see our response above.

Reviewer #2 (Remarks to the Author):

In this manuscript by Park et al., the authors train a neural net that reports on diffusion coefficients, throughout the field-of-view, from single molecule diffusion movies. The main novelty seems to be the use of a neural net on single-image motion blur to extract D per x-y bin. Overall the method seems to produce reasonable results. I suggest to address the following points to improve the manuscript. The main points relate to clarity of representation, comparison with SOTA, and incorporating temporal information.

Response: We thank the reviewer for his/her concise summary of our work and kind comments. Please see our point-by-point response below.

1. There is one basic question that comes up while reading the paper – even if there is information in a single molecule frame on D (and thanks to motion blur, this is a very reasonable assumption), wouldn't it make sense to also use the trajectory of the molecule? E.g. the distribution of spatial steps? (this point would come up in the next questions as well).

Response: Thank you for this discussion. Traditional single-molecule diffusivity analysis tracks every single molecule over many frames, which seriously limits the number of molecules that can be tracked in the sample, making mapping difficult. Thus, in this work, with Pix2D, we eliminate the need to track molecules, but directly extract D from single-molecule images captured in single frames. The reviewer brings up an interesting point that additional improvements might be achieved by further integrating single-molecule trajectories, but the difficulty is that for the SLB system examined in this work, as well as many other SMLM systems, each single molecule only shows up in a few frames (see our new Supplementary fig. 5). These short trajectories of different lengths complicate the setting up of models. Consequently, in this work we focus on examining how well we can extract D from single-molecule images captured in single frames, and leave possible further integration with molecular trajectories to future efforts. See also our new discussions. Page 6, Paragraph 2: “We also performed single-particle tracking^{12,13} on the data, but found most molecules only stayed in the SLB for a few frames (Supplementary fig. 5), as is typical in PAINT-type SMLM⁴⁹” Page 8, Paragraph 4: “While in this work we have focused on the direct extraction of diffusivity from single-molecule images in individual frames, future efforts may consider connecting molecules spanning consecutive frames to further enhance prediction.”

2. Unfortunately, there is no comparison with existing methods. For example – what does MSD or similar analysis on the experimental data look like?

Response: Thank you for this discussion. As noted above, our system is characterized by short trajectories (Supplementary fig. 5), making traditional MSD analysis difficult. Nonetheless, we have examined the occasionally observed long trajectories for MSD analysis (Supplementary fig. 6). We thus show that for the 3 SLBs of different compositions shown in Figure 3, MSD obtained comparable results as Pix2D, showing the same trend of decreasing D for the 3 SLBs. However, the low count of long trajectories disallows D spatial mapping. We have added related discussion to text (Page 6, Paragraph 2): “Mean squared displacement (MSD) analysis on the occasionally observed long traces (Supplementary fig. 6) yielded the same trends for the three SLBs...”.

3. To clarify – does each stack corresponds to a trajectory of a single molecule?

Response: As discussed above, in Pix2D, we do not track any molecules. Instead, each stack corresponds to different single molecules that are detected within the same spatial bin. These molecules are uncorrelated in time: they stochastically entered and left the SLB due to equilibrium with the aqueous buffer phase. To further underline the above point, we have revised the text to say “stacks of uncorrelated single-molecule images” in several places.

4. For each stack – the assumption is that there is no photobleaching of the molecule?

Response: As noted, these are independent events of different molecules stochastically entering and leaving the SLB. Photobleaching is not a major issue since the molecules are quickly replaced/exchanged through equilibrium with the aqueous buffer phase.

5. “molecules with their nearest neighbor closer than the signal sampling size were discarded”. What is the definition of “sampling size”? does that mean 2 molecules in the same 7×7 bin?

Response: For each localized molecule, we sample an ROI of 7×7 pixels around its center. When the ROIs of two molecules overlap, both molecules are discarded. Thus, none of the sampled ROI has an overlap. We have improved our description (Page 4, Paragraph 2): “A region of interest (ROI) of 7×7 pixels was sampled around the center of each localized molecule, and molecules with overlapping ROIs were discarded.”

6. What does “randomized sampling” mean (Fig 1g)? are the different frames within a single trajectory being “mixed”? is this what “permutations” (Fig 1 caption) refers to? Again, this is confusing because obviously there is information in the temporal order of the frames.

Response: As noted above, we do not track molecules, so the single-molecule images used are independent of each other. Thus, 100 inputs are generated from randomly sampling single-molecule images localized in each given spatial bin. Since single-molecule images in the same bin are independent (they are not the same molecule and are uncorrelated in time), the diffusion coefficient is not related to the order in the stack.

7. If I understand correctly, n_{ch} is the number of frames per trajectory that is being used. If so, I suggest to replace the word “channels” with “frames”, which to me seems clearer.

Response: Since these are not trajectories and we do not use any frame information, we have kept the term ‘channel’, which is commonly used for input dimensions in computer vision.

8. “For every bin satisfying $p_i > 10$, $m = 100$ datasets, each with the same dimensionality as the training data – thus m different $7 \times 7 \times n_{ch}$ arrays – were generated by random sampling from the p_i images for feeding into the model (Fig. 1g).”

Just to be clear - Does this mean that if a bin has, for example, only 11 images of a single molecule, these 11 images are replicated, in random permutations, to generate 4000 ($m=100$ times $n_{ch}=40$) images, which are fed to the net m times in batches of size n_{ch} ?

Response: Yes, as described in our Methods (Page 9, Paragraph 3), “For bins having $p^i < n_{ch}$, each input is generated by allowing repeated sampling of each image...” We find this approach robust and gives a straightforward solution to spatial bins containing varying numbers of single-molecule images.

9. Please add more detail on the final “regression layer” in the network.

Response: Thank you for this suggestion. We have provided more detail in the revised manuscript (Page 9, Paragraph 3 and Caption of Supplementary Fig. 1): “The final prediction was computed by a fully connected layer, followed by a regression layer that calculated the MSE loss between the predicted values and the ground truth labels.”

10. It seems (Supp figure 1) that there are no skip connections, and the network is not very deep, therefore citing “Resnet” might not be the best fit.

Response: Indeed, our network does not include the residual connections and is not as deep, which are two main contributions of ResNet. Nevertheless, we cite ResNet as an inspiration for our architecture due to its layer-aggregation strategy and the design methodology for deep neural networks. Please see our revised discussion (Page 3, Paragraph 3): “Whereas ResNet uses the ReLU⁴² activation function, we employed the Swish activation function for our regression task. A relatively shallow CNN model was implemented to avoid overfitting.”

11. Fig 2 c – the red line is not dashed, unlike what the caption says.

Response: Thank you for noting this. We have corrected it.

12. The SLB experimental results are said to be “consistent with the notion that diffusion is slower in more densely packed phases”, but there is no quantitative comparison to past results. Can this be obtained?

Response: We have improved related discussions (Page 6, Paragraph 2): “These values are comparable to previously reported results, in which $D \sim 3\text{-}4$, $\sim 2\text{-}3$, and $\sim 1 \mu\text{m}^2/\text{s}$ are reported for similar DOPC, DOPC:bSM, and DOPC:bSM:cholesterol SLBs at room temperature^{47,48}.”

13. In the SLB experiment - How long does a typical single molecule trajectory last before bleaching?

Response: Thank you for this discussion. We now include a new Supplementary fig. 5 to show the trajectory lengths. As the dye molecules reversibly enter and leave the SLB, they typically stay in the SLB for only a few frames, which limits the trajectory lengths as opposed to bleaching. As noted above, such short trajectories are difficult for MSD analysis but ideal for Pix2D.

14. Figure 4d is not clear – what does “single-molecule counts” (x-axis) mean, vs. “Counts” in the color-bar?

Response: Thank you for this comment. The “Counts” in the color-bar refers to the number of spatial bins that had the corresponding number of single-molecule counts and diffusion coefficients. We have revised this figure to clarify it, so the color bar says “number of spatial bins”. We also improved the caption to say “Two-dimensional distribution of the Pix2D-evaluated D versus single-molecule count for the different spatial bins...”

15. Regarding future work - It seems to me that the most immediate extension of this work would be to combine the motion blur information with step-distance information, could the authors elaborate on what that would require?

Response: As discussed above Re #1, indeed, even as in this work we focus on examining how well we can extract D from single-molecule images captured in single frames, integration with molecular trajectories could be an interesting future direction. We have added this discussion to our text (Page 8, Paragraph 4): “While in this work we have focused on the direct extraction of diffusivity from single-molecule images in individual frames, future efforts may consider connecting molecules spanning consecutive frames to further enhance prediction.”

Reviewers' comments:

Reviewer #1 (Remarks to the Author):

This work presents a machine learning (ML) method able to compose an spatial mapping of the diffusion coefficient of particles tracked by means of single-molecule localization microscopy. It presents both benchmarking results on simulated data as well as the application of the method to various experimental datasets. In general, I find the paper well written and convincing in its arguments. The method presented is very well presented and could be easily reproduced. Moreover, the authors provide Matlab codes for such. To note, the work builds strongly upon a method previously proposed by some of the authors (Ref 14), but extends it by means of ML techniques.

Here are some comments about the text. Of most importance are those related to the technical presentation of the ML method. I think that, after considering these, the work is suitable for publication.

1. The introduction covers very well the recent advances in the field, both in terms of the experimental approaches to study diffusion in biological environments, as well as numerical methods to assess their diffusive properties. The authors reference many works using ML in single molecule microscopy. However, I find that many of these references are mixed even if they treat different problems (e.g. localization, tracking, characterization of diffusive properties,...). It may be helpful for the reader to extend that sentence and separate these different applications. In particular, for the latter (characterizing anomalous diffusion with ML), I would suggest more recent references, e.g. [A] collects and benchmarks multiple methods as the ones referenced in Refs. 35 and 36. Moreover, [B] and [C] perform similar spatial maps and may be worth highlighting them.

2. Both in the abstract and the conclusion it is stated that "...Pix2D exploits the often undesired yet evident motion blur...". However, I can't find any proof backing up this argument in the results. I do not say that the CNN is learning to get information from this, but how can the authors be sure?

3. I also have concerns with the following, present too in the abstract and conclusion: "... the stochastic nature of diffusion mandates that even for a fixed D value, the motion pattern of a molecule in a fixed time window takes diverse forms not unique to the given D ". While I more or less understand the message the authors want to give, I think it is theoretically wrong. Given a D , the motion of a particle is fully characterized by such value. Having a short sample of the trajectory causes having poor statistics and maybe incorrectly predicting D . But these are not 'diverse motion patterns'. I would suggest the authors reviewing such sentences both in the abstract and in the conclusion.

4. The authors claim that ReLU activation is unsuitable for regression tasks. That is too strong of a claim, which should either be backed up by a proper reference or softened. In general, we can agree that other activations will work better, but that does not make ReLU unsuitable.

5. The authors express that "Training (Fig. 1d) was performed for each diffusivity label...". This seems to imply that different models were trained for different labels, which I guess is not the case? Also, it is not clear in the text that the diffusivity label equals the diffusion coefficient.

6. My major concern is the presentation of the training set and the definitions of p_i , m . I would suggest the authors to improve their description of these parameters.

7. Partially due to the latter, it is not clear to me what the size of the training dataset is. The authors state that the datasets are smaller than CIFAR and ImageNet, but having a number would be more useful to reproduce the work. Also, does the accuracy of the method improve by growing the training dataset?.

8. Could the authors extend on why they do evaluation only with $D_{\max} = 5$ but for instance do consider always the same D_{\min} ?

9. In Fig. 2b: i) how is the % Error calculated? Does a positive value mean overestimation or the opposite? In any case, the method does seem to over/sub estimate the error in some cases (e.g. $D = 4$). Can the authors comment on this?

10. It is not clear what happens when two particles are found in the same crop. It is said that for the benchmark "...molecules with their nearest neighbor closer than the signal-sampling size were discarded...". First, how is signal-sampling defined? Does this mean that only one particle will be found in each cropped image? This does not seem completely realistic for some of the experimental datasets considered. How can you deal with this?

11. The work considers that the spatial maps are static, but in such environments we can expect them to change. Have the authors considered such phenomena? Moreover, the authors consider that the diffusive properties emerge from the environment. However, there are many examples in which these changes occur because of the internal state of the particles. Is there any way of studying this effect with the presented method?

[A] "Objective comparison of methods to decode anomalous diffusion." Nature communications 12.1 (2021): 1-16.

[B] "InferenceMAP: mapping of single-molecule dynamics with Bayesian inference." Nature methods 12.7 (2015): 594-595.

[C] "Geometric deep learning reveals the spatiotemporal fingerprint of microscopic motion." arXiv preprint arXiv:2202.06355 (2022).

Reviewer #2 (Remarks to the Author):

In this manuscript by Park et al., the authors train a neural net that reports on diffusion coefficients, throughout the field-of-view, from single molecule diffusion movies. The main novelty seems to be the use of a neural net on single-image motion blur to extract D per x - y bin. Overall the method seems to produce reasonable results. I suggest to address the following points to improve the manuscript. The main points relate to clarity of representation, comparison with SOTA, and incorporating temporal information.

1. There is one basic question that comes up while reading the paper – even if there is information in a single molecule frame on D (and thanks to motion blur, this is a very reasonable assumption), wouldn't it make sense to also use the trajectory of the molecule? E.g. the distribution of spatial steps? (this point would come up in the next questions as well).

2. Unfortunately, there is no comparison with existing methods. For example – what does MSD or similar analysis on the experimental data look like?

3. To clarify – does each stack corresponds to a trajectory of a single molecule?

4. For each stack – the assumption is that there is no photobleaching of the molecule?

5. "molecules with their nearest neighbor closer than the signal sampling size were discarded". What is the definition of "sampling size"? does that mean 2 molecules in the same 7×7 bin?

6. What does "randomized sampling" mean (Fig 1g)? are the different frames within a single trajectory being "mixed"? is this what "permutations" (Fig 1 caption) refers to? Again, this is confusing because obviously there is information in the temporal order of the frames.

7. If I understand correctly, n_{ch} is the number of frames per trajectory that is being used. If so, I suggest to replace the word "channels" with "frames", which to me seems clearer.

8. "For every bin satisfying $p_i > 10$, $m = 100$ datasets, each with the same dimensionality as the training data – thus m different $7 \times 7 \times n_{ch}$ arrays – were generated by random sampling from the p_i

images for feeding into the model (Fig. 1g)."

Just to be clear - Does this mean that if a bin has, for example, only 11 images of a single molecule, these 11 images are replicated, in random permutations, to generate 4000 ($m=100$ times $n_{ch}=40$) images, which are fed to the net m times in batches of size n_{ch} ?

9. Please add more detail on the final "regression layer" in the network.

10. It seems (Supp figure 1) that there are no skip connections, and the network is not very deep, therefore citing "Resnet" might not be the best fit.

11. Fig 2 c - the red line is not dashed, unlike what the caption says.

12. The SLB experimental results are said to be "consistent with the notion that diffusion is slower in more densely packed phases", but there is no quantitative comparison to past results. Can this be obtained?

13. In the SLB experiment - How long does a typical single molecule trajectory last before bleaching?

14. Figure 4d is not clear - what does "single-molecule counts" (x-axis) mean, vs. "Counts" in the color-bar?

15. Regarding future work - It seems to me that the most immediate extension of this work would be to combine the motion blur information with step-distance information, could the authors elaborate on what that would require?